# The Ecumene: A Research Program for Future Knowledge and Governance

Paulo Castro Seixas [1,2] and Nadine Lobner [1,*]

1 Centre for Public Administration and Public Policies (CAPP), Institute of Social and Political Sciences (ISCSP), University of Lisbon, 1300-663 Lisboa, Portugal; pseixas@iscsp.ulisboa.pt
2 Centro de Investigação em Antropologia e Saúde (CIAS), University of Coimbra, 3004-531 Coimbra, Portugal
* Correspondence: lobnernadine0@gmail.com

**Definition:** The ecumene defines a beyond-border space of strong cultural encounters, flows, and merging, grounded within the traditions of world-systems, globalization, transnationalism, and cosmopolitanism discourses. Furthermore, the ecumene links directly with international regions as core political platforms in the making. As such, there are several ecumenes on the forefront, as evidenced by the literature, which can be clustered into ideal types. Epistemologically, it is a relevant concept and tool for a science-of-the-future that focuses on conviviality and transformation for the yet-to-come. Analytically, the ecumene has a descriptive, normative, and critical dimension, and can be empirically accessed through operational concepts such as triggers, hubs, and types of beyond-border conviviality. To apply the ecumene as a research program means to detect convivial common-sense spaces within the global context.

**Keywords:** ecumene concept; ecumene and future studies; epistemological turn; governance and trends of practices; beyond-border conviviality

## 1. Introduction

Humanity is facing a tipping point in which extreme scenarios are on the horizon. On one side, we see world culture on the rise; on the other, total fragmentation. This work is an attempt to extend science and knowledge, at large, as an active asset to cope with these "extreme scenarios". For this endeavor, we present the ecumene as a useful knowledge and political tool. Hence, we elaborate on the ecumene and ecumene studies as an innovative avenue of research for anthropology and the broader social sciences. We question: why the ecumene? If conviviality at large is a wicked problem, to tackle this, the ecumene could be the chosen concept. This exploration requires an epistemological reflection emphasizing a crucial turn in the production and horizons of knowledge.

Methodologically, this entry starts with a reflective analysis on the epistemological turn in which humanity is embedded in, supposing that we can already recognize certain possibilities of the new paradigm. We proceed with a historical and explorative literature review of the ecumene as a concept that can make a difference. Furthermore, we propose a specific research program for ecumene studies. Within this avenue, we discuss what the concept is about; what the grounds and empirical evidence through which scientists can expand their research are; the main dimensions that support the operationalization of the ecumene; and the methodologies and demarche processes that allow for the social construction of evidence.

The contribution of this work is to present the ecumene and ecumene studies, supported by an awareness of the future as the main cause of our present in a new design of knowledge. We consider that there is a turn from a probability evidence-based approach to a future test possibilities-based approach through which science may be understood as a continuous adjustment to turbulent waters. We elaborate on the ecumene as a concept (as a beyond-border space of strong cultural exchange, flow, and encounter) and ecumene

studies, through a descriptive, critical, and normative gaze, which are a way to match the current turn for the making (discovering/construction) of new common senses.

The paper is structured into the following sections: First, the epistemological turn will be explored, looking at the possibilities of a new paradigm. Thereafter, a discussion will be provided on the historical path of science to make sense of human togetherness through applying different concepts, such as society, culture, civilization, and politics, through which we further question their limitations. This will be screened through regions and hubs of future beyond-border convivialities and the role of anthropology to implement these new possibilities. Connecting to that, we will present an exploratory literature review of the concept and evident trends of practices addressing ecumene(s) in specific branches. This allows for a deeper debate on the wide-reaching potential of the ecumene in order for it to be applied as a research program in the future. Finally, this will be advanced through the elaboration on common-sense spaces within the world at large, through empirical evidence on one side and ideal types of conviviality on the other. It needs to be noted that this work stands in direct relation to an ongoing PhD project, "Objectifications, Networks and Worldviews of the Ecumene: the building of the EU and ASEAN as international regions in the making" where data are being collected through a multi-sited patchwork ethnography within center–periphery case studies (Lobner 2021–2024, supervised by Paulo Castro Seixas, FCT, 2020.07467.BD).

## 2. The Epistemological Turn

Knowledge production found itself within a paradigm transition in the 1960s, with a peak resulting from telecommunication, transport, and digital revolutions by the turn of the century (time and space compression). In 2024, it is assumed that light could be shed on some outputs of this paradigm shift, which will be the central point of this section.

As we are talking about a more-or-less 60-year time span, it is almost a human lifetime—or three generations. There have been a variety of shifts, and it is not possible to tackle them all. Core examples that can be mentioned are as follows: in this period, we have experienced a society shift (e.g., youth as wide-reaching social reality), a state shift (e.g., more than 100 new nation states), an international relations shift (e.g., relevance of international regions), a technological shift (e.g., digital and transport revolutions), and an ideological shift (e.g., digitalism as a way of life). Knowledge, at large, is, for sure, a main element within this turmoil and turn. One of the ways of looking at these shifts is through a tale that we may refer to as "Taking off the layers of the onion".

This is a story that serves as a useful metaphor for what is at stake. Here, the onion represents the truth of reality, and, therefore, is achievable through a rationality of evidence: the logical layer. Nevertheless, as Karl Popper argued, no matter how much evidence there is, we will never achieve the truth. The better possibility would be to identify hypotheses as false: the principle of 'falsifiability'. Hence, the accumulation of evidence is just what we have—not truth as the destination, but the probability of truth. This means that no hypothesis is ever completely proved right (although we are always accumulating evidence), but any hypothesis can, at any given moment, be considered false due to new evidence.

The second layer is the sociological one, considering the assertion 'the onion is the truth of reality' as a social construction. Thomas Kuhn [1] considered that truth is basically solidarity: a consensus amongst the scientific community about what truth is, and proper instruments in order to achieve it, i.e., 'the concept of paradigm'. This means that a paradigm represents a model and a way of thought of a particular time, space, and people, meaning that all knowledge is socially embedded and that it needs to be distinguished between a steady moment and a transformational moment (what Kuhn referred to as normal vs. extraordinary science).

The third layer is the psychological one, presenting that assertion as a possible plural one. The assumption is that there are either several truths about one onion, or about several onions. Paul Feyerabend [2] considered that individuality should be the focus. Hence, truth,

in addition to logic and sociological construction, is also an individual one. This means that a new gaze, mobilized by intuition and obsession, may present a new world and a new truth: 'the anarchist'/'the everything goes' principle. This is an open door to acknowledge the realities that were created by charisma or dictators, as well as to acknowledge that every individual counts for the acceptance of a plurality of realities/truths.

To summarize, this story is, in fact, the deconstruction of truth and reality as something imposed on us. Consequently, truth, reality and the future are an open ground for which a new framework is needed (see Table 1). To quote the final dialogue of the trilogy 'Back to the Future':

"Jennifer: Doctor Brown, I brought this note back from the future and...now it's erased! Doc: Of course it's erased!

Jennifer: But what does that mean?

Doc: It means that your future hasn't been written yet. No one's has. Your future is whatever you make it. So, make it a good one. Both of you!" [3].

The presented metaphor basically reminds us that transformation and reflexivity on transformation are the core of knowledge as an understanding of human society. Truth and reality are not something steady, but fluid: a journey. As such, the challenge is not just *to explain the boat*, but, moreover, a continuous *adjustment to turbulent waters*.

Therefore, we are facing a shift from a probability evidence-based approach to a future test possibilities-based approach. It is on this ground that we should question this new paradigm: what its main issues are; the assumptions and contexts of its social constructions; what the togetherness is that we are trying to build along with the definitions and accesses to it; and what its space/time frameworks are, along with its techniques and results. Following, a Table 1 is presented exploring the shift of paradigms.

If science becomes a set of test narratives in a more exploratory way (as, in a certain sense, it has always been), the challenge, in terms of understanding human society, is to test anticipatory conceptualizations. That is why, through this encyclopedia article, we present the ecumene narrative as (1) beyond-border conviviality in the making; and (2) as opportunity, possibility, and test on science, society, and politics reflexivity gaming.

**Table 1.** From PEBA to FTPBA: the adjustment to turbulent waters that we describe in this study.

| | **Probability Evidence-Based Approach (PEBA)** | **Future Test Possibilities-Based Approach (FTPBA)** |
|---|---|---|
| Main issue/agenda | Knowing (hypothesis, fieldwork, evidence) | Gaming (matchmaking, bets/tests and levels) |
| Main assumption | Escaping the unknown; Divided (subject-object) world | Awareness about the unknown unknowns; Holographic world(s) |
| Context/social construction | Disciplinary science; Authoritarian science | Sustainability science; Citizen/collaborative science |
| Definition of truth/target | Truth as probability: In a world of certainties, statistics give security | Truth/target as possibility: In a world of uncertainties and of unknowns, statistics and other similar evidence are not sufficient. Statistical outliers should be understood as possibilities/opportunities |

**Table 1.** *Cont.*

|  | **Probability Evidence-Based Approach (PEBA)** | **Future Test Possibilities-Based Approach (FTPBA)** |
| --- | --- | --- |
| Access to truth/ target | Paradigm [1]: Need for consensus; truth is solidarity | Opportunity for intuition [2]: Need for communities of practice; acceptance of difference; science as exercise to shed light on different possibilities; truth/target is opportunity |
| Time/process | Past and present; Feedback; History | Present and future; Feedforward; Utopia |
| Techniques/new methods/new path (how and why to perform it) | Descriptive and explanatory | Exploratory and testing |
| Results/contribution | Evidence narratives; Evidence-based policy; | Normative narratives; Policy-based evidence |

Source: Elaboration of the authors.

## 3. Does the Ecumene Make a Difference?

The use of specific concepts like society and culture, politics and civilization were attempts to identify a common world. These categories are, at the very basis of the reflexive quest of human encounter, for creating differences and/or commonalities. In the late 20th century, these concepts were not sufficiently explanatory anymore. Meanwhile, they were deconstructed in their coloniality. Other concepts—globalization, cosmopolitanism, transnationalism, world culture, etc.—emerged eventually, with similar issues. They are partly contents, and partly a genealogy of descriptive–normative–critical meanings that the concept of the ecumene encompasses.

In science and philosophy at large, there is the issue of creating concepts that could understand/explain similarities within the differences of 'humanity'. Ethnocentrism (the construction of the self against the other) has been the pattern for a long time, and, as such, humanity is a recent concept, consolidated only in the 18th century by the encyclopedists. Within humanity, in the last 200 years there have been several attempts to conceptualize similarities: society, culture, civilization, and so forth. Society and culture, along with politics and civilization, are examples of constructs/concepts which basically had the aim of both establishing limits and 'going beyond'. This means society and culture expressed a surpassing of biology and its social expressions within kinship (clans, lineages, and families). Hence, these concepts (at least in modern times) aimed to identify something larger that connects people/peoples beyond their internal differences. Usually, society and culture have been tied to the modern state, although with an essential tension between one and several nations (e.g., empires). Politics and civilization, eventually, are the concepts through which it was attempted to go further, connecting societies and cultures to a particular socio-spatial framework, i.e., cities and their increasing hegemonic power. This is why civilization—different from society and culture—is an ongoing process.

In the late 20th century, humanity reached a peak in which all major cities of the world became a network. Therefore, civilization turned out to be the recognition of this network: a center–periphery/capital–work struggle hierarchy; a financial capitalist hierarchy; a network of information flows; and a network-of-networks. Regarding politics, the 20th century also revealed the several "worlds" and their interrelations (first world, second world, third world, and even a fourth world). These world(s) politics and their rise and fall framed the entire planet eventually, for the first time, conveying politics as a never-ending dynamic of worlds in the making.

In the 21st century, the concepts of culture and society still do not have a settled meaning because of their implosion and explosion in subcultures and supra/super-cultures. As such, these concepts can be applied in a wide range of contexts (e.g., from world culture to community culture; from world society to street corner society; etc.). Initially (in the 19th century), they were used as a function of people (primordialism) and politics (modernism), either to support empire building, or to support state- and nation-building [4].

In present times, social sciences are not necessarily bound anymore by the politics of the nation state in both the conceptualization and operationalization of culture and society. This means that these concepts are liberated from the ascribed role (and obligation) of scientific legitimation of the nation, the state, or the empire. For this reason, nowadays, concepts such as pop culture, in many cases, go beyond the limitation of these former political categories.

The term civilization, in the English language, first appeared in the 16th century in the blooming of modernity, connecting the citizen (civis), the city (civitas), and the civil (civilis). To put it simply, in some moments of the historical path of social sciences, 'civilization' has been used to define 'living in cities' [5]. Civilization distinguishes human collectives through different forms of settlements, livelihood, economic systems, organizational patterns of societies, and others. Furthermore, it usually also refers to politics: how the occupation of an urban space and urban revolutions correlate with the formation of the state [6,7]. Therefore, "civilization", in close relation to "politics", supported certain conceptual distinctions within conflictual narratives between the state and other complex social hierarchies: historical vs. non-historical societies (hot and cold societies); print culture vs. oral culture; modernity vs. tradition; urban vs. rural; metropoles vs. colonies; and others.

The ecumene as a proposition goes beyond these differences. State and nation building, although still relevant (more so within postcolonial nations), paved and are paving the way to the building of international regions. Meanwhile, cities and their networks open(ed) up to city-regions and metropolitan and meta-metropolitan areas. Mobility and speed blurred and, in some cases, even erased the differences between the rural and the urban [5,8]. Other dichotomies such as the colony and the empire, indigenous and civilized, and tradition and modernity, are being criticized, and eventually erased. At the same time, gender, age, ethnicity, nationality and other categories are also being contested. As a result, new groundbreaking, wider framed categories are needed that encompass the critics and contestations, instead of merely being part of the problem. This is where, and why, the ecumene becomes of relevance for future debates.

*Regions and Hubs of Future Beyond-Border Conviviality*

Globalization, in the sense of the mundialization of economy, is a multi-secular trend. The Eurocentric perspective proposes that it had its roots in 15th century Venice, followed by Lisbon, Seville, Amsterdam, London, and finally, New York in the 20th century [9]. Nevertheless, as a result of transport and communication revolutions in the 1980s, the 'compression of time and space' [10] created the idea and practice of a 'global village' [11]. Simultaneously, hierarchy was smoothly substituted by networks and the space of the nation state by beyond-border relations [12]. Therefore, the core conflicts of the nation state (between capital and work) could be seen, by now, in a more encompassing framework: a center, a periphery, and a semi-periphery of the world-system [13]. Meanwhile, mundialization and globalization were concepts still attached to the first paradigm, and, as such, embedded mainly within descriptive and explanatory propositions. Despite this, reflexive analysis regarding globalization in the late 90s and the first decade of the 21st century presented already normative possibilities as evidence of the application of the second paradigm. This is the case with concepts such as alter-globalization, as well as globalization from below or globalized localism. Ulrich Beck's *risk society, subpolitics,* and *metamorphosis* are further evidence of this [14]. These concepts are already within the framework of the second paradigm, presenting a way to sail in turbulent waters. They are

proof of a bet in transformation, and not just descriptive and explanatory. They already represent possibilities and opportunities and, as a result, are part of a gaming framework.

The discussion that transnationalism, world culture, and cosmopolitanism bring is already much more open to possibilities than just a descriptive and explanatory position regarding existing societies. Transnationalism is rooted in studies about migratory flows and generations. Many studies link directly with double-bind national belonging, cross-cutting issues of coloniality, and empire. The world of diasporas and counter-diasporas, internal dislocates, and refugees that has been built in the late 20th and early 21st century resulted in a complex 'beyond-borders cultural soup'. Concepts such as creolization, hybridization, heterogenization, scapes, and flows are all manifestations of this beyond-border challenge. Consequently, transnationalism, more than a descriptive concept, is a range of possibilities.

World culture and cosmopolitanism, along with other similar concepts (e.g., global ecumene), present a horizon of expectations, where an open world of possibilities—utopias and teleologies—is needed. In this sense, ideas such as global civil society, global citizenship, global governance, world culture, etc. are ways to sail the boat in turbulent waters, which means to create directions or courses for identified possibilities. Therefore, transnationalism and cosmopolitanism are formulated in a much more open way than it was with the case of the mundialization of economy and globalization. The ideas expressed by those concepts are, moreover, normative, analyzing where the beyond-border possibilities and their patterns occur is, in itself, a test of and for the future.

It is within this framework that we propose the ecumene as a concept. The ecumene stands for an inhabited space where strong socio-cultural flows, encounters, and exchanges exist/take place. These areas/regions are grounds in which the future is played and tested by the people (bottom-up and top-down). This is why anthropology should focus on ecumenes: regions where we can detect the making of new forms of human beyond-border conviviality, in core hubs of such flows, encounters, and exchange. Different to globalization (used in a rather vague way), ecumenes do not necessarily stand for the world as a whole. Globalization was created in the idea of a unipolar world where the United States was the leading country. Consequently, globalization was often synonymous with Americanization [15]. Nowadays we live in a world that is widely considered multipolar (globalizations of fear and hope), and, for that reason, globalization is not (at least in singular) sufficient anymore. By using 'ecumene' as it is defined above, the focus is set on world regions where strong cultural exchange, encounter, and flow is taking place. On the other hand, transnationalism tackled, many times, phenomena that were fragmented and localized. Ecumene, in a rather different way, intends to identify the relations or narratives that create possibilities of materialities, networks, and worldviews that address world regions. It is within these regions that we should discover core hubs in which these encounters and exchanges create possibilities of future togetherness/conviviality.

Hence, we question, what is the role of the anthropologist regarding these hubs? There may be two alternatives: (1) the anthropologist who describes, analyzes, and creates explanatory models; and (2) the anthropologist with open eyes to anticipatory futures. For sure, both demarches are relevant, yet, for the purpose of this proposal, we emphasize the latter. In this case, the question is how some realities and perspectives are eventually more anticipatory than others, more likely as possibilities. We are aiming to understand where these grounds reveal triggers for new types of conviviality, picking up hubs where they are being active. Bearing in mind that there is an essential tension between possibilities/opportunities and horizons of expectations/utopias (Warren Wagar's "The City of Man" [16] is another relevant reference that offers significant historical and philosophical analysis of the idea of utopia and its meanings throughout human civilization. Wagar explores various utopian visions from ancient times to the modern era, examining how they reflect societal values, aspirations, and critiques of existing social orders), anthropologists and social scientists in general should be aware of this tension. Therefore, understanding

where the possibilities are is also a way to enable an anastrophe level that allows us to escape catastrophe [17].

In the next section we will present the state-of-the-art of the ecumene as a concept, and, thereafter, elaborate on its definition and layers.

## 4. The State-of-the-Art and Trends of Practice of the Ecumene

The exercise of this section aims at a state-of-the-art and exploratory literature review by screening the ecumene concept through its main categories and dimensions, which come to light through its applications over the course of time, and across various disciplines and thinking strands.

In historical and etymological terms, the ecumene first appeared in Greek designations (3000 BC) as the 'oikoumene' (οἰκουμένη), referring to something 'inhabited', hence, the world as an inhabited space of human beings. It descends from the 'oixus' (inhabited; house) and the 'nenon' (space) (e.g., [18–21]). Therefore, the initial purpose of the ecumene was to characterize a common ground (eventually, a joint household) of human interaction and co- and intra-habitation. Being on the very core of the foundation of the Roman Empire (31 BC), the concept became a way to characterize civilizational processes and constitutions, as well as the division between, for instance, the sacred and the profane imperial administration. In light of this, the ecumene became a main reference for the Christian church in order to designate the assembly of bishops all over the world and their 'unified whole' as being a relevant part of the 'civilized world' (the Roman Empire). Constantinople, the capital of the Roman Byzantine Empire (330–395 AC), was termed the 'Ecumenical City', whereafter its ruler was named the 'the Ecumenic Patriarch of Constantinople' [21].

Even today, both within the sacred and profane/administration dimensions, the role of the border frontier contact zones in a division-and-bridging way may be identified as a layer of the ecumene concept. To put this under the lens of the 21st century, religions continue to use the ecumene to refer to bridging different belief systems—the ecumenic dialogue. In the case of border contact zones, Jerusalem is a very relevant case, as it is being claimed by three different religions: Judaism, Christianity, and Islam. Istanbul, too, serves as an intriguing case of observation—a city in-between civilizations, a gate between the western and eastern worlds, as well as the bordering between Europe and the Arab world (and beyond). Nevertheless, the literature on the ecumene is rather scarce and can be divided into six main dimensions, as visible in Table 2.

**Table 2.** Preliminary exploratory literature review.

| Ecumene Dimensions in Contemporary Literature | | | | | |
|---|---|---|---|---|---|
| *Linguistic* | *Juridical* | *Political* | *Historical/Geographic* | *Socio-cultural* | *Religious* |
| Sanskrit Ecumene [22] Ecumene of Languages [23,24] Esperanto in Portugal [23] | Bioethics Ecumene [25] Human Rights Ecumene [26] | Ecumene during Holocaust [27] Indian Trade Ecumene [22] Commodity Ecumene [28] | Changing World History [29] Ecumenopolis [30] Visions of the Global Past [31] Ottoman Ecumene [32] Mesoamerican Ecumene [33] | National Cultures in Global Ecumene [34] Global Ecumene [20] Caribbean Oikomene [35] Lusotopy as Ecumene [21] Global Lineages [36,37] | Roman Ecumene [38] Buddhist Ecumene [39] Islamic Ecumene [40–42] Christian Ecumene [43] |

Source: elaboration of the authors.

Certainly, through an exploratory scientific literature review, similar phenomena may be subsumed in different conceptualizations. There are other concepts which, at least partly, have similar meanings and are evidence of the same quest. As mentioned in earlier steps of this work, society, culture, civilization and politics may sometimes be used with the same intentions and meanings. Also, in the case of the "interculturalism branch", we can find relevant concepts such as, for instance, communities of communication [44] and diatopical hermeneutics [45]. In the area of cognitive anthropology, the concept "modes of thought"

is particularly relevant as a way to create human similarities and translations. Hence, we are facing an open narrative which requires constant further analysis and reflections.

The categories reached through the exploratory literature review evidence dimensions that strongly interrelate with the leading categories we have presented in the previous section. What correlates in all is the reference of having a certain common denominator that helps to create a 'bond' between participants within the respective beyond-border realms. We may take a closer look into how this is presented within the exploratory literature review, as described in Table 2. Each section reveals a certain reference point, stretching from language to law, politics to history, and socio-culture to religion. Each author included within the table uses the ecumene to frame a shared space beyond borders. The literature to be found in the JSTOR and Web of Science databases clarify that either historical–geographic, socio-cultural, or religious explorations are considered through the ecumene. By presenting this literature review, the concern is how the concept is displayed in scientific and contemporary discourses. Hence, we will zoom into the debates that aim at making sense of the ecumene as a concept.

Strongly embedded within a primary Christian context in latter years of the 20th century, the ecumene turned into a universal 'vision' of the Christian church, the togetherness and cooperation of a variety of Christian communities: 'Ecumenism' [43]. The core target of Ecumenism was to provide a ground for negotiations between several Christian groupings, strands, and inter-group reflections, as well as the mixing of rituals and visions. The ecumene is still used in a very frequent manner under the spelling of 'oikoumene' in the Christian context for promoting interdenominational cooperation and liaison. The World Council of Churches, for instance, named its official website after the concept [46,47]. Hence, within religious debates, we found different avenues that relate with the ecumene as a concept: ecumenism, understood as a unity among Christian faith communities [34]; the acknowledgement of several religious ecumenes (Christian ecumene; Islamic ecumene; Buddhist ecumene; etc.); and a global interreligious dialogue, for instance by advocating for diatopical hermeneutics [45].

Considering this historical development and religious usage of the ecumene, a new dimension has been added in the 20th century in scientific discourses within political debates, philosophy, and geography. The first time the ecumene was used to describe bigger complexities within social scientific approaches was by Lewis Mumford in order to debate the rise and influence of 'technique' onto civilizational processes—and vice versa [48]. Furthermore, another strong political debate that makes use of the ecumene concept was published in 1963 by William McNeill, in which he firstly proposes the 'global ecumene' to be a result of the boom of late modernity, predominantly visible through (back then) internationally dominating European political institutions, rising scientific discourses, technology, and extreme economic expansion [49]. Parallel to that, a number of discourses about the colonial framework, as well as former geographical debates, make use of the ecumene to distinguish the 'old world' from the 'new world' (colonizers vs. colonized), leading to the consideration of two different ecumenes, which seem to have merged after the imperial–colonial 'discoveries'/encounters [50]. The idea of civilization as a world of interconnected cities was always subliminal to the concept of the ecumene. Doxiadis, a Greek architect and urbanist [30], proposed Ecumenopolis as a unified global city, constituted by a network of several cities.

Concerning social sciences, one of the first influential and descriptive ideas of the ecumene within our field was undertaken by Alfred Kroeber back in 1945. Kroeber explored the concept as a tool/output for increasing cross-border interlinkage mechanisms, acknowledging that the world continues to grow in its connectedness at large. Through a culture-relativist gaze, he applied the ecumene to precisely tackle transnational connections through a diffusionist approach. Coming from a historical pathway, he tried to understand and describe humankind as an 'interwoven set of happenings and products' [19]. His focus (considering the historical context he was embedded in) was to tackle sociocultural diffusion over the planet. Even though Kroeber argues from a cultural diffusionist position,

some of his ideas remain in present debates when talking about the ecumene, especially pertaining the interwoven cultural and social aspects spread over the world. Kroeber understood very early that there is a key connector between humans, making them 'relatives' in continuous translation [19] in times when such ideas were most neglected. Some decades after Kroeber, Ulf Hannerz made further use of the concept for understanding transnational movements and interconnectedness [20]. Hannerz talked about a 'network of networks', in which he proposed the global ecumene to be an international realm of human interconnection that went beyond common ideas of space and time. His idea of the ecumene aimed to overcome arbitrarily constructed divisions in order to cope with inevitable global human (inter)activity. This opened up rich avenues of social flows beyond geographically proximate borders, a product of historical networks, i.e., movements of people of the (colonial) past based on political and economic interests between the center and the periphery.

Shortly after Hannerz' application of the ecumene, in 1996, Sidney Mintz used the concept to come closer to understanding the cultural interconnectedness of the Caribbean region [35]. Based on the indicator of creolization, Mintz debated the regional togetherness of a strong diversity of populations within the Caribbean, analyzed through a historical Marxist perspective. Bringing large-scale modification to the forefront, which was activated through the colonial presence and its mechanisms, through the Caribbean Oikoumene, Mintz argued that the region has been on the very core of modernization dynamics reinforced through industrialization, slavery, and large-scale manufacturing for European commercial needs. Hence, in this case, the regional togetherness of this precise area is described as an ecumene based on a clear historical, shared pathway of former power and exploitation structures and thereof resulting coping strategies by the region's inhabitants. Back in the time of the first colonial encounters in the region, populations from various parts of the world were brought to the islands for establishing a strong labor force through slavery. Each social group that was brought to the region was of different origin. Considering this specific form of forced migration to the Caribbean region, its broad variety of populations formed their new common identity, which persists through Caribbean descendants [35]. This 'Caribbean oikoumenê' is therefore based on shared feelings of the past through which a new togetherness rises as a form of internalized resistance against a common enemy, which has been the set of European colonization throughout several centuries. To summarize, it can be said that Mintz' main argument is that the ecumene becomes evident through the strong intercommunication beyond national borders, which is, in this case, more intense due to a shared past (The concept of new societies is a relevant link to this outline and should be deeper explored in the continuance of ecumene research. Portugal, for instance, within its expansion period in colonial times, created 'new societies', of which literature remains scarce. Madeira island was, presumably, the first imagination of a 'new society' (a mix between people from Africa and Europe) as a first human experiment in such terms. The most important example, though, is Cape Verde, shortly before the large 'experiment' of human mingling in Brazil. The 'Estado Novo' concept of 'Lusotropicalism' evidences a precise ideology of such supposedly new societies [51]).

When looking at this outline of the ecumene in a regional context (located on a recognizable 'united' geographical area—in the sense of mapping), we may move further to another exploration of the ecumene by João Pina-Cabral, who explores lusotopy within this framework supported by the idea of a historical and intersubjective reality construction in a global context [21]. This work tackles a more complex dimension of human interconnectedness beyond borders than geographically proximate countries within a specific region. The Caribbean, opposed to the abstract definition of 'lusotopy', is rather 'easy' to spot on a practical world map. Lusotopy, as an ecumene, needs—in geographical, historical, as well as socio-cultural terms—a variety of other attentions in order to be grasped. As Pina-Cabral moves further within this realm, he pinpoints that it cannot be 'located' or separated through national borders, language, or the past colonial empire. It cannot be 'broken down' into CPLP (Community of Portuguese Language Countries), for instance, and also not into

Portuguese diasporas. More than such 'clearly defined' aspects, the writer proposes that lusotopy is based on the complex dynamics of interweaving processes, hence, an ecumene: he considers a togetherness of Portuguese links and interconnections all over the world, such as sharing a common sense beyond cultural patterns, language, social ties: in short, amity/kinship. Despite including the obvious facts of the togetherness of Portuguese speaking countries, the historical past and spaces with strong Portuguese habitation as a result of demanding and strongly impacting global occasions (e.g.: Portuguese Jewish exiles), and Portuguese working migrations (Switzerland, France, etc.), the author suggests that—in this case—the main point of a common ground beyond recognizable borders is the 'amity' established between such complicated pasts, presents and the building of future(s). Using amity and, in certain cases, kinship as relevant indicators of this ecumene model, Pina-Cabral considers 'spaces' rather than 'countries' when talking about lusotopy within this framework [21]. What remains at the very core is the great importance of a certain familiarity and common sense between people/persons sharing 'space' on several levels.

What can be taken from this ecumene model is that the interweaving, fluid, and transformative dimensions of space and time are core drivers of human interconnectedness in a world-at-large context, deeply considering interpersonal and intersubjective modes of identification. The ecumene is therefore a catalyst for the building of an international space of strong cultural exchange, based upon emotional bonds and proximity deeply embedded within codependent reality construction.

Another interesting, yet rather historical–geographic/religious, interpretation of the ecumene is to be found in the text 'Beyond the World-System: A Buddhist Ecumene' [39], in which the author considers a historical Buddhist ecumene prevailing. Goh focuses on a 'geospatial religious and political subsystem that existed between Buddhist common-wealth of world system'. Here, the ecumene serves to understand a cultural and political-religious network (back in the 11th to the 14th century) that had the aim of creating a world ruled by a universalist monarch. Hence, in the 11th century, new interstate relations were established between Myanmar, Thailand and Sri Lanka based on this idea of a Buddhist Ecumene. The main argument is that the Buddhist Ecumene was applied for a back-then geographical 'supraregion' within three Buddhist centers, strongly influenced by common religious and textual patterns, with the aim to be led by a universal world conquering monarch.

Also relevant for this debate, Christian Reus-Smit [26] goes beyond 'traditional' ideas of international relations through his positioning of the ecumene within the discipline. As he critiques the limitations within past/current IR discourses towards global matters, he emphasizes the need of anthropological and sociological knowledge expansion within the rather pragmatic approaches of IR. He published a work in which he tackled the occurrence of the ecumene in an international order setting: global human rights. Whilst elaborating on the quest of international/universal human rights, Reus-Smit explains the 'ecumenical space' as 'marked by multiple forms of modernity, by products of interaction between western cultural ideas and practices and other civilizational complexes' [26] (p. 1209). Through connecting the idea of the ecumene to an institutional, juridical and, at the same time, ethical quest, he states that 'individual rights' need to be acknowledged as the political core of the global ecumene. He uses the ecumene to analyze complexities of cultural politics of the globalization of human rights, international institutional contexts, and the relevance of global cultural diversity. The author argues that there is a strong need to establish a bridge between disciplinary and institutional strands, of which a possible case could be global human rights, for which he strongly emphasizes the importance of debating and acknowledging the global ecumene (For further exploration, see Boaventura Sousa Santos [52] on cultural translation between Sharia and Western law, "Toward A Multicultural Conception of Human Rights").

In 2020, we undertook an intense exploration of the ecumene for beyond-border discourses with the case of Timor-Leste and its role on the world stage [36,37]. We used the ecumene to understand bottom-up constructions of Timorese people, through which we reached a set of ideal types. These ideal types help us understand the role of the

ecumene for building a shared human ground and to enforce conviviality through social (local) perceptions about the world at large. Timor-Leste, a small country with rather weak international representation, proved to be a relevant case to elaborate on this domain through its global interconnectedness to political, economic, as well as socio-cultural 'players': regional actors such as the EU, ASEAN, CPLP, commonwealth, etc. Through testing the ecumene within this avenue of international relations vs. anthropology, we understood the inevitable importance of opening the debates to construct a more profitable, translatable future of humanity.

In addition to the literature review and the conceptual universe that evidences the awareness and focus on beyond-border relations as well as the ideal types, we can already refer evident trends of practices addressing ecumene(s) in specific branches. The following table presents an exploration of trends of practices without an exhaustive and systematic attempt that should be addressed in future analyses. Nevertheless, this attempt to systematize the practices towards ecumenes is an effort in which some organizations are already engaged as well.

As demonstrated through Table 3, there are already certain ecumene trends of practices that should be taken into account. Our attempt to cluster these follows main categories of togetherness within the narrative of modernity. Certainly, there is a need for new categories in a new world. Elsewhere [16], we proposed that rationalities of an ecumene(s) world would focus on two main branches: ecumene studies (broadening human togetherness) and sustainability studies (broadening human-nature togetherness).

**Table 3.** Ecumene trends of practices.

| Religion | Economy and Politics | Culture and Society | Ethics and Legal Norms |
|---|---|---|---|
| Ecumenism; interreligious dialogue; world parliament of religions, 1893; diatopic hermeneutics; etc. | The League of Nations; United Nations; the World Constitution; International regions such as EU and ASEAN; etc. | International NGOs (science, women, environment, sports, humanitarian/human rights); global civil society organizations; Esperanto, 1887; English as lingua; etc. franca; FIFA (211 countries); etc. | Legal pluralism (idea and practices); commission for legal pluralism; international commission of jurists; association of lawyers for democracy and human rights; permaculture ethics; UN agendas (MDG and SDG); etc. |

Source: Elaboration of the authors.

A gaze from a distance [53] and to identify and create world(s) that surpass borders have been a human aim ever since. A system cannot be analyzed in its completeness and consistency without a perspective from a broader one [21,54]. Within this path, religion was the first attempt to have a gaze from a distance, whereas *modernities* present a diversity of attempts. This is the case of "society and culture", "ethics and law", "economy and politics". They are examples of the modernity of enlightenment application of science as an instrument of the human reason/human mind considered as one.

The broader perspective of the community yet to come is a responsibility of everyone, for which this paper aims to be an elaborated contribution. For proceeding with this matter, the next section will present the ecumene as a research program.

## 5. Ecumene: A Research Program

Historically, the ecumene has had several understandings and applications, but its core objective has always remained the same: to detect and promote a shared space of mutual understanding in a beyond-border realm. We further argue that the ecumene can be understood as a descriptive, critical and normative concept:

*i.* Triggers and hubs of beyond-border conviviality, to be found in shared spaces of strong cultural flows, encounters and exchange (descriptive definition), in which it is possible to discover;

*ii.* A togetherness of people from different socio-economic classes, from different socio-cultural bounds (e.g., ethnicities, nationality, gender), and with different ecological backgrounds and expectations (social critique);

*iii.* The manifested or latent goal of peaceful and secure human common sense (normative description).

Supported by this, our purpose is to identify the proper contexts in which an ethnography of triggers, hubs and types of beyond-border conviviality is applicable. Our hypothesis is that the framework of international regions in the making as shared spaces of strong cultural flows, encounters, and exchanges are rich grounds to implement a program of ecumene ethnographies. We believe that international regions are platforms of triggers and hubs for the overlapping and overcoming of inequalities in which new forms of conviviality as future opportunities emerge.

As demonstrated in the previous section, by the end of the 20th and beginning of the 21st century, the ecumene offered new potential in scientific discourses, precisely within social sciences by the discovering and construction of a ground of mutual understanding through specific ideal type conceptualizations: creolization [20,35]; amity [21]; and global lineages [36]. These ideal types appear to be led by a broad question, which also guides our proposed research program: How can the ecumene be used to make sense of bottom-up/top-down interconnections within the World at Large?

Firstly, the ecumene is manifested through two main axes, space and time. On one side, the ecumene refers to a beyond-border conviviality. On the other side, it can be considered that these types of conviviality are guided by constant attempts to build a shared future. Secondly, the ecumene may be grasped within macro- and microdynamics. As the ecumene is to be understood as a space of strong intercultural encounter and exchange, international regions (as well as country-regions) in the making are platforms for this beyond-border conviviality (macro-dynamics). Furthermore, within meso-dynamics, cities (as well as other glocal spots: in some cases, they are digital (social media-scapes), in some cases ephemeral (transformational festivals), and in some cases, sites with ancient relevance (Angkor Wat, Kathmandu, etc.)) are crucial observation platforms for grasping where a broadening of strangeness (Georg Simmel, in his essay "The Stranger" [55] he proposes *strangeness* as the relation of the big urban city. In the 21st century, the cities created spots that enhanced the broadening of this *strangeness* relationship (airports, intermodal centers, university campus, heritage centers, sports centers, thematic parks, etc.)) takes place. In microdynamics, types of conviviality (conjunctive and disjunctive) are the specific subject of research. They are empirical facts that should be described and analyzed through a critical gaze. Furthermore, they may serve as social and political tools and as such, can be explored by applied anthropology. Hence, there is a descriptive, critical, and normative dimension. Thirdly, an empirical demarche on the ecumene may be enveloped both by scientific concepts, as well as by operational concepts. Regarding the former, we consider that the dimensions of material representations, social networks, and worldviews are relevant. In the case of the latter, hubs, types of conviviality and triggers are at the core. This will be further elaborated in the next paragraphs, presented in a systematic way (Table 3).

Material representations, social networks, and worldviews are three dimensions that help to guide the empirical exploration of the ecumene. They provide deep insights into how beyond-border relations are constructed and maintained and expressed within city brokerage platforms. Cities, in this context, are seen as centers for constructing new units through the cross-over of the before and the after, towards new transformational settings (through the meeting of several meaning systems, human extensions, and axis-mundi cities). Hence, by focusing on the three mentioned research dimensions, the understanding of beyond-border conviviality through flows, encounters, and exchanges (in a symbiotic top-down/bottom-up domain), resulting in the building of international regions, will be enabled.

A research program on the several dimensions may be realized as follows: cultural mapping and/or a categorization of ecumene objectifications of spaces of strong cultural

exchange (e.g., museums, monuments, sites, building, etc.); sociograms built from people's flows within international region contexts as evidence for shared beyond-border meaning systems (e.g., transnational students; expats; digital nomads; lifestyle travelers); and shared worldviews (e.g., transformation; environmentalism; mobility freedom).

We propose that these dimensions (objectifications, networks and worldviews) enhanced by third space actors (with "third space actors" we refer to brokerage social agents which are in-between spaces, such as countries, ethnicities and others [56]) and relations that are supported/understood by different ecumene ideal types (e.g., creolization, amity, global lineages, transnational student/expat cosmopolitanism, and identities from the margins). This approach aims to support the understanding and awareness of different layers and cross-overs of beyond-border conviviality and togetherness; hence, different ecumenes in the making. Furthermore, the three dimensions are embedded within main (reciprocal) scientific axes: 'space', through an 'anthropology of beyond borders' tackling globalization, transnationalism, and cosmopolitanism (e.g., [20,57–59]; and 'time', through an 'anthropology of the future', concerning the tensions within the discipline between the origins and conviviality, focusing on the yet-to-come (e.g., [60,61]).

Therefore, the aim is to tackle the anthropological/social science shift from the past to understand the various constructions and perspectives of and towards the future in order to build a shared roadmap for coping with former and current global clashes (In the course of a contemporary PhD thesis project (Lobner 2021–2024), this is currently in process with the EU and ASEAN as case studies with a specified center–periphery focus: Brussels and Lisbon in the EU; Jakarta and Hanoi in ASEAN). Regarding empirical indicators we considered three main ones: triggers, hubs and types of conviviality. Triggers are a collection of elements/traces that acquire the attention of the ethnographer for a beyond-border "social situation" (as elaborated and applied by the Manchester school by, for instance, Max Gluckman and Clyde Mitchell, but also Clifford Geertz). As such, we are referring to an open list that always needs to be updated. By hubs, we mean centers in which new beyond-border common senses emerge. For now, resulting from the exploratory work completed, we distinguish between two main hubs: (1) hubs that are part of the ecumene paradigm culture, and (2) hubs that are part of specific city cultures (replicable formats). Regarding types of conviviality, we refer to the ones in which beyond-border relations are constitutional and contractual and the ones in which to participate in beyond-border relations are flexible and situational options. In both cases, we will have conjunctive and disjunctive conviviality, as conflict is always a possibility.

This categorization has, for us, a similar rational as the ideal types of Ferdinand Tönnies [62], when he tried to tackle "the world of yesterday" vs. "the world of today" by Gemeinschaft-Gesellschaft, through kinship and status vs. interests and contract. Another rationale may be followed by the concept of "communitas" by Victor Turner [63]. As in the examples given by Turner, we propose that contemporary beyond-border conviviality in the making already grounds specific communities. These are in opposition to the regulatory and conjunctural beyond-border relations supported by a world of nation states. We consider materialities, gathering opportunities, and worldview possibilities as triggers for beyond-border conviviality, which will be demonstrated in Table 4.

**Table 4.** Ecumene dimensions and empirical indicators.

| Empirical Indicators | Hubs | Conviviality | Triggers |
| --- | --- | --- | --- |
| **Dimensions** | | | |
| Material Representations Social Networks Worldviews | Ecumene Paradigm Culture (e.g., Hostels; Festivals) Ecumene City Culture (e.g., Café Culture Brussels; National Theater Brussels; Superblock Culture Jakarta; etc.) | Conviviality grounded by beyond-border relations (communitas, e.g., communities of international artists, ecovillages) Beyond-border relations as conjunctural conviviality (flexible and situational, e.g., Erasmus, subcultural events, sports) | Materialities (Hubs, Sites, Clubs, Equipment of the ecumene city culture) Gathering Opportunities (Surf Culture, Rave Culture, Climbing Culture, InterNations events, online apps) Worldview possibilities (desire for connection, need for tran sformation, dissatisfaction with the system) |

Source: Elaboration of the authors.

These dimensions and empirical indicators, meanwhile, may open up to explore the ecumene in three different definitions, as referred previously by a descriptive, a critical, and a normative gaze.

i.    Descriptive

The concept of the ecumene proposed in this text refers to a test/bet in possibilities of hope for broader strangeness engagement by top-down or bottom-up agencies (airports, scientific and technological parks, international sports training centers, cultural heritage sites, hostels, festivals, etc.). It is a ground of mutual understanding, where certain denominators can be detected which connect the participating entities. These convivialities, as empirical facts, should be described, categorized, and measured. The ecumene is not a singular phenomenon; more than that, we argue that there are several ecumenes on the world stage which eventually help to create common sense between human beings across the planet.

ii.    Critical

In modernity, three critical movements are seen to be entangled: the first one focuses on socio-economic inequality, considered as a social construction. This awareness implies an alternative horizon of expectations (e.g., communism) to overcome the existing situation. The second critical movement, which was emphasized in the latter decades of the 20th century, focuses on an awareness of a plurality and fragmentation of inequalities (e.g., gender, age, ethnic groups, race, and so forth). The third critical movement brings human vs. nature to attention, in which nature is the subaltern. Therefore, the ecumene should be understood through the theories of conflict within capitalist, patriarchal, colonialist, and anthropocenic hegemonies, among others. In these three movements, there is an awareness of the divisions, and, implicitly, the idea of overcoming them. Several authors have, in the past, tackled this avenue by proposing different categories/scenarios: awareness of the world as a system or biosystem [13,64–66]; the realistic utopia of life of dignity and emancipation for all [67]; homogenization/indigenization [54]; homogenous world, complete fragmentation, and a world of possible translations [68]; creolization; and reaction to diversities [20] (These references, certainly, are not exhaustive but the aim is to give a significant evidence of a specific tradition).

Understanding that conflict and dissensions are crucial in a convivial world, the ecumene further presents several levels of human disagreements, which bring a desire and need of cultural translation to the frontstage. Human conviviality may be conjunctive or disjunctive; therefore, there is an interplay between consensus and contestation, for which a middle ground needs to be generated/detected.

iii.    Normative

Following the critical perspective, ecumene can be a tool and a desire. It is a way to actively reflect on the future and implement transformation. One of the meanings of modernity is the unfinished ground of thoughts and works of humanity considered as a whole because the world is still not just one or the unlimited community of communication has still not been reached [43]. This modernity mobilized several possibilities of instrumentalization of reason (following the enlightenment legacy and the encyclopedia example) [67]. In the late 1980s (with the example of Tony Blair and Anthony Giddens), the discursive reconfiguration of the cultural industries concept of the Frankfurt school paved the way for a positive gaze regarding instrumentalization of reason. This turn eventually has been a trigger to global pop cultures.

Late modernity, postmodernity, and other post-eras are ways to deal with the essential tension between the acceptance of diversity and the need for teleology [58]. Eventually, to depict several ecumenes and possible middle-ground translation, dispositives may be considered as configuration of possibilities within this dilemma. We need to question how these three definitions serve to explore ecumene(s) in a more elaborate way. For such an attempt, we propose two methodological possibilities for closer observation: a bibliographical state of art on the ecumene and a presentation and analysis of existing ideal types.

*The Ecumene for a World of Conviviality: Detecting Spaces of Common Senses within the World at Large*

The aim of this text was to bring the ecumene to the forefront of scientific debates, highlighting its relevance to make sense of the world at large. But how can the ecumene be detected for understanding beyond-border interconnections for the building of a convivial human future? As mentioned earlier in this paper, our statement is that the ecumene can be observed in a dualistic interconnected way for translating its various layers:

(1)    Empirical evidence—positivist-phenomenological way

One possibility to observe the prevailing of different ecumenes is through quantitative data. We understand it as relevant to observe phenomena in order to obtain the greater picture of how movements, flows, and telecoupling occur in the world at large context (The current research focuses on the EU and ASEAN as ecumenes in the making, through the earlier mentioned ongoing and funded PhD project (FCT, University of Lisbon, Lobner 2021–2024)). Through this gaze, it is possible to collect factual, testable information about the world. It allows for the gathering of data that prove inter- and cross-regional movements, helping to visualize dynamics of strong human exchange. This, we propose, can be gathered through quantitative data of interactive world maps which reveal movement/migration charts, air traffic data, data on the use of the internet/social media, and others. In addition to quantitative attempts, there is a phenomenological qualitative analysis, focusing on triggers, hubs and types of conviviality within the three dimensions material representations, social networks, worldviews. The fluidity of bottom-up dynamics should be centered in order to enhance the emergence of models that represent ecumenes in the making.

(2)    Ideal types of interculturality—interpretivist-normative way

The other relevant gaze for observing ecumenes is qualitative data that allow a prescriptive nature of the characteristics of interconnected populations. By the phenomenological approach referred above (hubs, types and triggers of conviviality through particular dimensions), ideal types are to be explored and expected. The literature review of the previous section offers some of the main analytical dimensions of the Ecumene uncover that a set of ideal types already prevailing (creolization, amity, and global lineages). This emphasized the complex domain of transnational/beyond-border constructions for creating human common senses. In order to advance research within this avenue, existing and new ideal types should be considered and may serve as a basis for conviviality policy making.

In both avenues—empirical evidence and ideal types—we consider digital-scapes and cities as contextual brokerage (Nevertheless, we need to consider (and further explore) that cities as brokerage places are not always just positive/beneficial. Sometimes they constitute borders between civilizations [69] and are continuous stages of brokerage through conflict and war) spaces that help to discover ecumenes in action, demonstrating strong intercultural exchange, opening a linkage between center and periphery, between the micro and the macro, between individuals and the world.

This gaze towards conviviality should be (beyond merely descriptive) critical and even normative. As understood by this exploration, strong conviviality should enhance the overcoming of the three generations of inequalities (economic, socio-cultural, human vs. nature). Therefore, finding these triggers, types and hubs of conviviality is a critical engagement in a positive gaze (Usually, a Marxist critical engagement trend focuses research on the inequalities. In many cases, this demarche only emphasizes inequalities without giving hints or clues for its overcoming. This is precisely where ecumene studies present a potential turning point. To focus instead on hubs of conviviality which already present the overcoming of several inequalities is an opportunity to showcase best practices to be replicated) which should highlight "best practices" to be replicated in a more widely way in order to broaden and to deepen the ecumenes. This is where ecumene studies pave the way to the future through a normative gaze. Finally, a relevant distinction needs to be made between past, present, and future. Certainly, there have been numerous ecumenes in the past which strongly impacted world dynamics (e.g., kingdoms; nation states; empires; etc.). Such past dynamics do, with no doubt, still influence present structures. Yet, we argue that there is inevitable necessity in grasping current ecumenes in the making, for the awareness about new futures which enable creating commonalities through differences. Nowadays, the challenge seems to be an awareness of international regions as realm where the experience of conviviality is stronger and therefore, the elected ground for social scientists to engage in the ecumene studies.

## 6. Conclusions

The debate that we enhanced within this work draws upon the ecumene and ecumene studies as constructive and innovative ways to cope with human togetherness from the 21st century onwards. We strongly elaborated on how to approach our joint future through a perspective that suggests a surpassing of the limitations of the past. This position paper considered that we are within an epistemological turn in which the concept of the ecumene makes a difference. The ecumene, understandable through triggers, hubs, and types of conviviality in strong cultural beyond-border flows, encounters, and merging, is explorable through a descriptive, critical, and normative dimension. For this purpose, we presented a research program for the broadening of knowledge and governance in a community/communities yet to come, within the world at large. Our aim was to contribute to a rethinking and deep reflection of contemporary social sciences, in order to jointly pave the way for a common human path towards a convivial future.

This is not a unique attempt merely to our interest. It is deeply embedded within contemporary efforts to test new common grounds and possibilities to overcome conflict, dissent and planetary destruction. This is, for instance, the case with different formal and informal networks that implement the ecumene for a common transformation agenda, such as the Ecumene Project and Ecumene Residence (ecumeneproject.com) that advocates for art as a shared experience and a universal dialogue to enrich communities around the world. Other attempts include the ecumene congress on sustainable development and COP28 (law research center, iclrc.ru); the ecumene discussion club as a platform for coping with economy, financial agendas, sustainability and development matters; and the ecumene journal of environment, culture and meaning (1994–2001). Also, similarly within this avenue, the World Values Survey (WVS) is a global research project that examines socio-cultural and political change over time. It surveys individuals in various countries to

understand their beliefs, values, and motivations as represented through a (cultural) map that categorizes countries based on their cultural values and attitudes.

These are examples of bottom-up and top-down, private and political, as well as civil society efforts to apply the ecumene within a joint global human agenda.

Therefore, by finishing this exploration, we, at the same time, want to open up for activating the ecumene in different impact zones: scientific programs and educational platforms; the implementation within the making of public policies and best practices; and humanitarian and environmental action. This, certainly, requires ongoing testing and gaming in a future-focused mindset, envisioning transformation and innovation at large.

**Author Contributions:** We state that the provided work has been of equal contribution. Conceptualization, methodology, investigation, formal analysis, writing–original draft preparation, and writing–review and editing has been undertaken equally by P.C.S. and N.L. The project has been administered and supervised by P.C.S. All authors have read and agreed to the published version of the manuscript.

**Funding:** This research was funded by FCT (Fundação para a Ciência e a Tecnologia), 2020.07467.BD; CAPP (Centro de Administração e Políticas Públicas), and CIAS (Centro de Investigação em Antropologia e Saúde).

**Acknowledgments:** This work is a direct output of the continuance of several international projects that helped to elaborate further on the ecumene, such as the EU H2020 Project CRISEA, a master thesis (Lobner 2020) supervised by Paulo Castro Seixas and Nuno Canas Mendes, and an ongoing FCT funded PhD thesis project under close supervision of and guidance by Paulo Castro Seixas. We want to deeply thank our colleagues from international relations, Nuno Canas Mendes and Andrea Valente, who, on a regular basis, bring their expertise and support to the evolvement of our work, and our interdisciplinary, international research team consisting of the authors of this paper and Ricardo Cunha Dias, Ines Subtil, and Diogo Guedes Vidal. We also thank our research centers and funding sources for the support of our work.

**Conflicts of Interest:** The authors declare no conflicts of interest. The funders had no role in the design of the study; in the collection, analyses, or interpretation of data; in the writing of the manuscript; or in the decision to publish the results.

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
