# Peer review of "The Ecumene: A Research Program for Future Knowledge and Governance"

_encyclopedia, doi:10.3390/encyclopedia4020051_

Round 1

Reviewer 1 Report

Comments and Suggestions for Authors

The article brings an interesting proposal about the "usefulness" or the "aim" of the Social Sciences. Faced with a historical vision in which they occupied legitimizing mechanisms of other entities (such as, for example, the Nation-State) and after the epistemological crisis which the Social Sciences are still living through, the article brings an interesting proposal.

The ecumene would be highly dynamic ambiguous and multisituated place through which empirical confronted with ideal types can be seen as a prospective play for constructing hypothecticxal broader sociabilities. It is a suggestive and quite daring proposal, but it is well founded and presented with rigor.

Reading becomes a little difficult at times due to its strongly theoretical nature, but some examples help to navigate it better. As a small criticism, I may say that the tables werea bit difficult to navigate.

Author Response

We deeply want to thank the reviewer for these constructive insights and elaboration.

We are delighted to see that the proposed theme is reaching the understanding of the reader in terms of significance. We are aware of the complexity of the topic, and we believe that it is of uttermost relevance to scientifically engage with the future.

We adapted the tables accordingly and hope that by now, they bring more clarity.

Reviewer 2 Report

Comments and Suggestions for Authors

My primary impression on reading through it is that it is obscurantist in relying intensively on a vocabulary that is barely elaborated and remains for the most part on the level of abstraction. The genealogy of the concept ecumene is scattered historically, and the 'leap' from religious conceptualisations to the 'new' forms is not particularly clear (the reference to Pina-Cabral's thoughts on Lusotopy provides a rare comprehensible insight into what the authors seek to define as ecumene). The use of reference suggests the authors have a strong familiarity with relevant texts but they rarely if ever expand on why those references are relevant; I cite here one salient example of the many cases of assertion without clear if any grounding -- "modernity mobilized several possibilities of instrumentalization of reason (following the enlightenment legacy and the encyclopedia example) (Adorno and Horkheimer 1944). In the late 80s (with the example of Tony Blair and Anthony Giddens), the discursive reconfiguration of the cultural industries concept of the Frankfurt school paved the way for a positive gaze regarding instrumentalization of reason. This turn eventually has been a trigger to global pop cultures." I found the tables and charts offered somewhat jejune and not helpful in clarifying the arguments. Finally, it's a bit disturbing that the lauded research on ecumenes cited at the close seems to be that of the authors of this paper.

Comments on the Quality of English Language

English is good although one wonders why the introduction opens in past tense.

Author Response

We confirm that we lean on abstraction, although with some clear practical, applied proposals, such as the ones of Joao Pina-Cabral, Sidney Mintz, Ulf Hannerz and our own research based on empirical fieldwork in Timor-Leste. In the case of the ecumene, which is still a theme in the making, we did our best to bring clarity into the given complexity.

As the concept itself has been used in such a variety of forms, and up until today not with a clear line, we had no other option than collecting the works which have been done and present it in their specific truths. This leads to the fact that sometimes, it can seem a little misleading due to what the reviewer states as “scattered”; nevertheless, this is the reality we need to face with existing literature and applications of the ecumene.

We are pleased to be acknowledged about the familiarity with the used references. As for the critique on the example given from our text, we would like to state that it was not the intention to deeply elaborate on every specific subject, but rather to give a sophisticated overview of the relevant layers. Regarding this precise quotation of our text, our intention was to understand Ecumene as one of the possible instrumentalizations of reason and the turn-arounds (negative and positive) of the process.

We re-read the text in order to try to be more clear in each case where such a doubt could occur.

We absolutely agree to the concern of the tables and charts, reviewed them accordingly in order to adapt them to a better understanding and perception. The research already done by the authors on the Ecumene, and as referred in the end of the paper, works as support research that offers empirical insights, and the primary motivation for the writing of the present paper. The paper at hand is a corollary of a research path, which is the main reason why we refer to our previous research. Furthermore, there are no other examples found that apply the ecumene in an empirical way, for which reason we hope it is understood that through this work, we propose a new line of research.

Finally, we deeply want to thank the reviewer for the careful and constructive revision, and the important suggestions placed.

We adapted the English accordingly.

Reviewer 3 Report

Comments and Suggestions for Authors

This is a very interesting essay about important issues in social science research. The arguement for employing the ecumene as an orienting concept is persuasively made but i do not think it applies to science in general but rather to the social science disciplines.  There are two target audiences for this essay:

1. humanity in general, and 2 , social sciences.  the goal of moving toward greater consensus about ontology and values is admirable and the review of literature on this topic is good. I can recommend two additional citations on this topic: The World Value Survey Cultural Map, and Warren Wagar's _The City of Man_.  It might also enhance the discussion to reflect more on generational issues.  The idea of multiple ecumenes is interesting. It might be a good idea to consider how different world civilizations have been converging and the extent to which ecumenes are nested into world regions, or cross-cutting (lusotopia).  And some discussion of emerging ideas such as white indigenism and white nationalism might also be included.   

Regarding the idea that social scientists should all move to the testing and community development turn, i think it makes more sense to legitimate the notion that this be an important new direction for the more activist-inclined

and policy social scientists.  it now reads as if the older weberian approach to objectivity should be completely abandoned.  

Comments on the Quality of English Language

english is good but some light copy-editing is needed without wrecking the poetic tone of the writing. 

also some citations are not in the references.

Author Response

We deeply want to thank the reviewer for this perception and understanding of the proposed theme.

We are also very grateful for the very useful literature recommendations, and implemented it into our work.

We agree to the reviewers reflection, which is completely aligned with the ideas expressed within the text.

This recommendation will be considered for further analysis. Although certainly relevant, it would need deeper argumentation that would suit the text at hand. Therefore, we are grateful for the references and will elaborate further on them in our upcoming works.

We believe that we joint the descriptive, the normative and the critical levels without declining any of those dimensions. As such, what we intended to present was that the positivist perspective should go together with the utopian one.

We tried to make sense of past-present-future dynamics. Nevertheless, we propose that the main triggers for present actions are future perspectives. As such, the positivist/objective analysis of the present is always needed, but the comparison should be more with the future narratives than with past narratives.

By re-reading the text we adapted to these suggestions.

Reviewer 4 Report

Comments and Suggestions for Authors

This is an interesting and very informative entry on Ecumene, a concept that finds descriptive, critical and normative dimensions.

I find well done the systematic review, and the organization of the many contributions and ideas around that concept.

Possibly the authors could consider some more texts, like David Krieger, "The New Universalism" (1991); indeed a consistent theological tradition has pointed insistently in that same direction since 3 decades.

My only question is to what extent social systems theory, like the one developed by Parsons, or later by Niklas Luhmann, and other close models later, (Pierre Bourdieu), could help to better elaborate that idea, based in the view ow World society as a global integrated system, sharing many commonalities and structures. This could work at least at the descriptive and perhaps, the critical levels.

Author Response

We deeply want to thank the reviewer for the careful engagement with our work, and we are delighted to see that it has reached the comprehension of the reviewer.

We deeply want to thank for the reference suggestions, which we have implemented.

We are grateful for the constructive insights on Krieger and will deeply consider it in further works. For the given paper, it would need stronger elaboration and therefore, will not be included.

We included the suggestions about the view on world society to the extent that was possible, as these are very helpful elaborations for the work at hand. Nevertheless, for deeper exploration we will need stronger engagement with the named topics in another work.

Round 2

Reviewer 3 Report

Comments and Suggestions for Authors incomplete reference: p 21 [69] Wagar fn 10 on p 19 "These references, certainly, are not exhausted,"  (exhaustive) the article still needs copy-editing for occasional English blunders.  Comments on the Quality of English Language   fn 10 on p 19 "These references, certainly, are not exhausted,"  (exhaustive) the article still needs copy-editing for occasional english blunders. 

Author Response

We deeply want to thank the reviewer for the second round of going through our work. We have reviewed the article again and were able to detect some more mistakes which are, by now, corrected. We hope that the work, with its current status, applies to the high quality standard of the journal.